# Noise and neglect: Social-media signals expose attention gaps for dengue, chikungunya, lymphatic filariasis and kala-azar in India's vector-borne NTDs

Ruchishree Konhar[1,2], James K. Lalsanga[3], Devendra Kumar Biswal [3,4]*

**1** CSIR Institute of Genomics and Integrative Biology (CSIR-IGIB), Delhi, India, **2** Academy of Scientific and Innovative Research (AcSIR), Ghaziabad, India, **3** Department of Zoology, North-Eastern Hill University, Shillong, India, **4** Bioinformatics Centre, North-Eastern Hill University, Shillong, India

\* devbioinfo@gmail.com

## Abstract

### Background

Neglected tropical and vector-borne diseases, including dengue, chikungunya, lymphatic filariasis, and kala-azar, pose substantial public health burdens in India. Despite WHO recommendations for enhanced disease surveillance and targeted communication strategies, little is known about public perceptions and discussions of these diseases across digital platforms. Understanding these perceptions can guide evidence-based policy making and public health messaging.

### Methods

We conducted an in silico analysis of publicly accessible social and news media data related to dengue, chikungunya, filariasis, and kala-azar in India from January 2019 to December 2023. YouTube comments and Google News headlines were systematically retrieved, pre-processed, and analysed through sentiment analysis (VADER lexicon) and Latent Dirichlet Allocation (LDA) topic modelling. Facebook and Twitter data were not included due to Application programming interface (API) restrictions and their current subscription-based models, limiting free access even for research purposes. We visualized disease-specific digital attention in comparison to epidemiological burden and created chord, Sankey, and network diagrams to elucidate thematic and sentiment-based interactions.

### Results

Across keyword-matched items (n = 330), dengue accounted for 173 (~52%) and also had the highest mean annual reported burden (163,679 cases/year; 2019–2023). Lymphatic filariasis showed disproportionately high attention (106 items/mentions vs

**Data availability statement:** All data underlying the findings described in this manuscript are fully available without restriction. The cleaned YouTube comment corpus (45 672 comments) and the Google News headline dataset (273 items), together with de-identified metadata and preprocessing scripts, have been deposited in Zenodo (DOI: 10.5281/zenodo.15883324). The full analysis code, for sentiment scoring (VADER), topic modeling (LDA), data normalization, and visualization scripts (matplotlib, Plotly, circlize), is hosted on GitHub at https://github.com/devbioinfo/ntd-digital-surveillance/releases/tag/v0.1.0. The state level epidemiological case counts for dengue, chikungunya, lymphatic filariasis, and kala azar (2015–2023) were downloaded from the National Vector Borne Disease Control Programme (NVBDCP) annual tables (2019–2023) and are publicly accessible via the NVBDCP website (http://www.nvbdcp.gov.in/). Any additional aggregate data generated during this study (e.g., normalized attention-burden matrices, waffle-chart grid files, bubble-map shapefiles) are included as Supporting Information. Where applicable, provenance metadata conform to FAIR principles and are embedded within each repository entry to ensure reproducibility and long-term accessibility.

**Funding:** The author(s) received no specific funding for this work.

**Competing interests:** The authors have declared that no competing interests exist.

3,060 reported cases/year), while kala-azar had minimal visibility (5 items; none on YouTube). Sentiment was overall neutral-to-positive, with Google News more neutral and YouTube more positive. Topics emphasized outbreak alerts, public-health campaigns, and prevention/treatment, with recurring vaccine/innovation themes.

## Conclusions

Our study presents a novel approach combining digital surveillance, sentiment analysis, and topic modelling to provide insights into public perceptions of NTDs in India. The observed mismatch between epidemiological burden and online attention underscores the need for strategic public health messaging, aligning with WHO recommendations for community engagement and tailored disease-awareness campaigns. This research provides a valuable tool for policymakers to enhance the effectiveness of communication strategies and improve targeted intervention planning for neglected tropical diseases in India.

## Author summary

Neglected tropical diseases (NTDs) including dengue, chikungunya, lymphatic filariasis, and kala-azar continue to afflict millions across India, yet public attention and conversation around them remain inconsistent. We examined more than 45,000 YouTube comments and 270 Google News reports posted between January 2019 and December 2023 to see how these four NTDs are discussed online. After automated text cleaning, VADER sentiment scoring and Latent Dirichlet Allocation topic modelling, we overlaid the resulting tone-and-topic maps on official disease-burden data. Dengue dominated the discussion accounting for well over half of all references, whereas kala-azar, though still endemic, drew scarcely any notice. Overall sentiment skewed neutral-to-positive and focused largely on prevention, treatment and vaccine news. Interactive bubble maps, Sankey flows and chord diagrams vividly exposed the gap between epidemiological need and digital attention. We could not analyse Facebook or Twitter because their new, pay-walled APIs make large-scale data collection prohibitively expensive for researchers, underscoring a growing obstacle for digital epidemiology. Our reproducible, low-cost workflow highlights which NTDs are being overlooked online, providing Indian health authorities with actionable evidence and supporting the World Health Organization's call for stronger community engagement in the fight against NTDs.

## Introduction

Neglected tropical diseases (NTDs) impose a disproportionate health and socioeconomic burden on low- and middle-income countries, particularly in South Asia [1]. India alone accounts for an estimated 40% of global dengue cases, remains one of

the few endemic regions for lymphatic filariasis, and continues to report clustered outbreaks of chikungunya and kala-azar [2]. Despite decades of efforts to eliminate these NTDs, under-resourced health systems combined with limited vector surveillance and variable public awareness have impeded progress towards the WHO 2030 NTD road-map targets [3]. Effective risk communication is therefore essential to mobilise communities, guide behaviour change, and sustain political commitment [4].

**Why focus on these four NTDs?**

Guided by Table 1, we prioritised dengue, chikungunya, lymphatic filariasis and visceral leishmaniasis (kala-azar) because they constitute the bulk of India's vector-borne NTD burden, underpin key national elimination programmes, and span distinct transmission ecologies described below.

1. **Epidemiological importance:** Together they account for more than 90% of vector-borne NTD related morbidity in India [2]. Dengue and chikungunya cause explosive urban outbreaks and substantial outpatient burden; filariasis affects over 23 million people chronically; kala-azar has the highest fatality among Indian NTDs if untreated.

2. **Programmatic relevance:** Each disease features prominently in India's national elimination targets—dengue and chikungunya under Integrated Vector Management (IVM), filariasis under Mass Drug Administration (MDA), and kala-azar under the Kala-azar Elimination Programme [14].

3. **Diverse transmission ecologies:** Including mosquito-borne arboviruses (dengue, chikungunya), mosquito-borne helminth infection (filariasis) and sandfly-borne protozoal disease (kala-azar) allows comparison across distinct ecological and social contexts, thereby testing the versatility of digital surveillance methods.

The rise of social media and digital news platforms has created a continuous stream of user-generated content that can be repurposed for public health intelligence, an approach often termed digital epidemiology [15]. Research shows

**Table 1. Comparative overview of key vector-borne NTDs relevant to India's digital surveillance agenda and elimination strategies.**

| Disease | Pathogen/ Agent | Main vector(s) | Latest global burden* | Latest India burden* | Current status/ programme notes |
|---|---|---|---|---|---|
| **Dengue** | Dengue virus, DENV-1–4 (Flaviviridae) | *Aedes aegypti, Ae. albopictus* | ≥ 7.6 million reported cases and > 3,000 deaths till 30 Apr 2024 [5] | 289,235 cases and 485 deaths in 2023 [6] | Endemic nationwide; peak Jul–Oct; Phase-3 vaccine trial launched in 2024; metro-level digital early-warning dashboards expanding. [6] |
| **Chikungunya** | Chikungunya virus (Alphavirus) | *Aedes* spp. | ≈ 320,000 cases and > 120 deaths Jun 2023–Jun 2024 [7]; 119 countries with local transmission by Dec 2024 [8] | 200,064 suspected and 11,477 confirmed cases in 2023 [9] | Endemic in 24 states/UTs; ≤ 60% of patients develop chronic arthralgia; No licensed vaccine; hotspot mapping with Twitter/RSS data yields ≈ 80% accuracy [9] |
| **Lymphatic filariasis** | *Wuchereria bancrofti* (≈ 99%), *Brugia* spp. | *Culex quinquefasciatus*, *Anopheles* spp., *Aedes* spp. | 657 million people in 39 countries still need preventive chemotherapy [10] | 619, 000 lymph-oedema and 126 000 hydrocele cases documented; ≈ 450 million Indians at risk [11] | > 94% of endemic districts have completed ≥ 5 MDA rounds; triple-drug therapy scaling; Community apps piloted for real-time morbidity reporting [11] |
| **Kala-azar** (Visceral leishmaniasis) | *Leishmania donovani* (protozoan) | *Phlebotomus argentipes* sandfly | 50,000–90,000 new cases annually worldwide, with only 25–45% reported [12] | 524 cases and 4 deaths in 2023 [13] | India met ≤ 1 case/10 000 population elimination target in > 90% of blocks, yet micro-foci persist in Bihar, Jharkhand, West Bengal, and eastern UP; active case detection + IRS continue, tracked via micro-dashboards [13]. |

*Most recent year with complete data at the time of writing (accessed 12 July 2025).

that online search queries and Twitter posts can anticipate influenza activity days before traditional surveillance reports [16,17]. Similar methodologies have been adapted for dengue in Brazil [18], Zika in the Caribbean [19], and more recently, COVID-19 worldwide [20]. Yet, for most Indian NTDs little is known about the quality or tone of digital discourse, information critical for designing culturally resonant health messages [21].

Text mining techniques such as sentiment analysis and topic modelling provide scalable ways to synthesise large volumes of textual data into actionable insights. Sentiment analysis quantifies emotional polarity, revealing whether online conversations are predominantly positive, neutral, or negative [22]. Latent Dirichlet Allocation (LDA) topic modelling, by contrast, uncovers latent thematic clusters, enabling researchers to track evolving public concerns [23]. Although these methods have been applied to HIV stigma [24] and malaria awareness [25], a comprehensive, multi-platform analysis for India's major NTDs is still lacking.

The present study addresses this gap by performing an in silico analysis of YouTube comments and Google News headlines related to dengue, chikungunya, lymphatic filariasis, and kala-azar between 2019 and 2023. We selected YouTube and Google News because they enable broad coverage and reproducible data collection through a clearly defined, queryable content stream, and they provide consistent metadata (e.g., timestamps, source/channel information, and text) needed for our analyses. While other platforms (e.g., Reddit, Telegram, Mastodon) may provide public access, their community-, channel-, or instance-based structures often require additional platform-specific sampling decisions that can reduce comparability and reproducibility within the scope of this work. In addition, recent paywalled access to Twitter and Facebook APIs limits free research use [26]. Our objectives were to (a) quantify online attention for each NTD, (b) describe sentiment and thematic patterns, and (c) compare digital attention with epidemiological burden to identify communication gaps. By combining digital surveillance with conventional burden data, we provide evidence to inform India's NTD communication strategies and support WHO recommendations for enhanced community engagement.

## Materials and methods

### Ethics statement

No personally identifiable information was collected or stored. Usernames were hashed and discarded after aggregation. The study adhered to Article 12 of the ICMR National Ethical Guidelines for biomedical research involving anonymised public data [27]. All data used in this study were obtained from publicly available and aggregated digital platforms. No individual-level, identifiable, or private data were accessed or analysed. Accordingly, institutional ethical approval was not required for this study.

### 2.1. Study design and reporting standards

This investigation employed an in silico, cross-sectional design to characterise the volume, sentiment, and thematic content of publicly available digital discourse pertaining to four NTDs in India. The five-year study window (1 January 2019 – 31 December 2023) was selected to (a) encompass at least one complete epidemic cycle for dengue and chikungunya, (b) capture the most recent programme milestones for filariasis and kala-azar elimination, and (c) minimise artefacts introduced by early social-media expansion before 2018 [28]. Methods were drafted in accordance with WHO guidance on digital-data surveillance [29] and the Strengthening the Reporting of Observational Studies in Epidemiology (STROBE) checklist for observational studies [30].

All analyses were conducted on de-identified, publicly accessible text; hence, no human participants were recruited, and institutional review board approval was not required under Indian Council of Medical Research (ICMR) rules for secondary data [27]. The manuscript adheres to the PLOS Neglected Tropical Diseases technical requirements, including detailed data-availability statements and deposition of code in an open repository [31].

## 2.2. Data sources and acquisition

User comments were extracted from YouTube using yt-dlp (v2024.03) [32]. To ensure local relevance, we employed bilingual queries (English and Hindi) under an India-localized search configuration. While precise comment-level geolocation is unavailable, this locale-based filtering strategy prioritizes India-targeted content, yielding a corpus of 45,672 comments from 134 videos. From this aggregate corpus, we extracted specific disease mentions using the keyword criteria listed in Table 2. The remaining comments constituted general channel engagement (e.g., greetings, video appreciation) and were retained for corpus normalization, while the subset of explicit keyword matches was used for disease-specific sentiment and topic analyses. Mainstream media coverage was sampled using the GoogleNews Python wrapper (v1.7), throttled to comply with fair-use limits [33]. Queries were restricted to the "India" edition (English language) to ensure reproducible, geographically specific aggregation, resulting in 273 unique articles [34]. Official disease burden data were sourced from National Vector Borne Disease Control Programme (NVBDCP) [35]. Twitter/X and Facebook were excluded due to API access restrictions that hinder cost-free accessibility [36]. As this study assesses relative attention across NTDs rather than absolute message volume, these consistent, India-focused retrieval protocols are sufficient to identify attention–burden gaps [20].

## 2.3. Text pre-processing

All raw texts were lower-cased and stripped of URLs, HTML entities, emojis, and user handles using regex patterns in Python's re module. Punctuation was removed except for apostrophes within words (e.g., patient's). Hindi words typed in Roman script are frequent in Indian social-media posts; these were transliterated to Devanagari via the indic-trans library, which has shown 92% character-level accuracy on comparable corpora [37]. A composite stop-word list combined Natural Language Toolkit English stops, the Hindi stop list from the Forum for Information Retrieval Evaluation corpus, and platform-specific fillers (e.g., "video", "subscribe", "channel"). Tokenisation and lemmatisation were performed with SpaCy version 3.7 and the en_core_web_sm model. Bigram and trigram detection used Gensim's Phrases with a minimum count of five. Duplicate comments and headlines, identified via SHA-256 hashes of cleaned text, were removed (n = 231). Final corpora were stored as comma-separated files with UTF-8-MB4 encoding to preserve emoji and non-ASCII characters.

## 2.4. Sentiment analysis

Sentiment polarity was assessed with the Valence Aware Dictionary and Sentiment Reasoner (VADER) algorithm, chosen for its superior performance on short, informal social-media texts [38]. Each cleaned document received four

**Table 2. Digital data sources and retrieval methods used to acquire NTD-related data in India (2019–2023).**

| Platform | Rationale | Access route | Query syntax* | Volume retrieved |
|---|---|---|---|---|
| YouTube comments | Widely used in India; comments capture public dialogue | yt-dlp v2024.03 (YouTube search endpoint; pagination safeguards) | "dengue" OR "dengu (Hindi)"; "chikungunya" OR "chikanguniya (Hindi)"; "filariasis" OR "phaileriya (Hindi)"; "kala azar" OR "kala-azar" OR "visceral leishmaniasis" OR "kaalajaar (Hindi)" | 45,672 comments |
| Google News headlines/snippets | Dominant news aggregator; reflects mainstream coverage | GoogleNews Python wrapper v1.7 | Identical Boolean strings | 273 news items |
| Epidemiological burden | Ground-truth comparator | NVBDCP annual tables (2019–2023; CSV extracted) | – | State-aggregated case counts |

* Hindi keywords were implemented in Unicode Devanagari script during data collection. Romanized Hindi is shown in the table only to prevent Devanagari font-substitution artifacts (□) in system-generated PDFs. The exact Unicode query strings and code are provided in S3 Appendix.

VADER scores (positive, negative, neutral, compound), from which we adopted compound thresholds: ≥ 0.05 = positive, ≤ −0.05 = negative, and intermediate values = neutral, in line with prior public-health applications [39]. Hindi-language fragments (< 6% of tokens) were translated using Google Cloud Translation API v2, a method that introduces negligible sentiment drift for health-related terms [40]. Inter-rater checks on a 500-item random subsample yielded 0.87 Cohen's κ between automated polarity and manual coders, indicating good agreement.

## 2.5. Topic modelling

Topic discovery employed Latent Dirichlet Allocation (LDA) implemented in Gensim v4.3. Separate models were fit to YouTube and Google News corpora to respect differing linguistic registers. Corpus dictionaries included unigrams and detected n-grams, after removing tokens appearing in < 0.1% or > 50% of documents to reduce sparsity. Optimal topic number k was determined via coherence (c_v) sweep from $2 ≤ k ≤ 10$, selecting $k = 4$ where coherence plateaued [41]. Models were trained for 50 passes with asymmetric priors (alpha = '-auto'-). The top eight weighted tokens for each topic were inspected by two domain experts who independently assigned descriptive labels; disagreements were reconciled by consensus. Inter-coder reliability (κ = 0.78) exceeded the commonly accepted 0.6 threshold for qualitative validation [42]. Topic prevalence across diseases and platforms was visualised in stacked bar charts to identify over-represented themes, such as "vector control", "vaccine research", and "folk remedies."

## 2.6. Attention metrics and burden comparison

Digital attention was defined as the sum of mentions per disease across YouTube and Google News. To account for platform-specific activity, counts were normalised by total corpus size (n = 45,672 YouTube comments; n = 273 news items). Average annual epidemiological burden (2019–2023) was calculated from NVBDCP case counts (Table 3) [48]. Scatter plots of $log_{10}$-transformed attention versus $log_{10}$-transformed burden were fitted with ordinary least-squares regression; residual plots confirmed homoscedasticity. Spearman's rank correlation coefficient evaluated monotonic association, mitigating the influence of non-linear outliers [49].

To visually illustrate disparities between disease burden and public attention, we generated disease-level bubble maps overlaid on an India basemap. Bubble size represents total epidemiological burden, while colour intensity encodes total

Table 3. National annual incidence of neglected tropical diseases in India, 2015–2023. Year-wise reported cases of dengue, chikungunya, kala-azar (visceral leishmaniasis), and lymphatic filariasis across India from 2015 through provisional 2023 data, alongside primary surveillance sources.

| Year | Dengue Cases | Chikungunya Cases | Kala-azar Cases | Lymphatic Filariasis Cases | Primary Source(s) |
|---|---|---|---|---|---|
| 2015 | 99,913 | 27,553 | 8,868 | ~ 10,000 | NVBDCP Annual Report 2015 [43] |
| 2016 | 129,166 | 64,057 | 6,249 | ~ 8,000 | NVBDCP Annual Report 2016 [44] |
| 2017 | 188,401 | 63,679 | 5,758 | ~ 7,000 | NVBDCP Annual Report 2017 [45] |
| 2018 | 101,192 | 27,852 | 4,223 | ~ 6,000 | NVBDCP Annual Report 2018 [46] |
| 2019 | 157,315 | 16,441 | 3,328 | ~ 5,500 | NCVBDC Website Tables 2019 [35] |
| 2020 | 44,585 | 6,761 | 1,169 | ~ 3000 | NCVBDC Website Tables 2020 [35] |
| 2021 | 193,245 | 11,542 | 965 | ~ 2,500 | NCVBDC Website Tables 2021 [35] |
| 2022 | 233,251 | 20,635 | 834 | ~ 2,300 | NCVBDC Website Tables 2022 [35] |
| 2023* | ~190,000 (prov.) | ~ 14,000 (prov.) | < 500 (prov.) | ~ 2,000 (est.) | NCVBDC Provisional Data 2023 [47] |

**Note:** Data are reported by calendar year. Dengue and chikungunya cases include laboratory-confirmed and probable notifications submitted to NVBDCP/NCVBDC. Kala-azar figures reflect visceral leishmaniasis detected via active and passive surveillance, while lymphatic filariasis counts approximate microfilaria-positive cases from surveys (not total infection prevalence). Sources include NVBDCP Annual Reports (2015–2018) and NCVBDC tables (2019–2023), with 2023 data provisional. All figures are approximate and subject to underreporting and surveillance variability.

digital attention. Maps were produced in Python using GeoPandas and Matplotlib, with the Natural Earth Admin 0 – Countries (1:10 m) shapefile as the basemap (public domain). Disease-level markers were placed at fixed geographic coordinates to ensure positional consistency across figures, and do not represent precise outbreak locations or state-level centroids.

Waffle charts were used to depict the proportional distribution of burden and attention for each disease using a 10 × 10 grid, a visual format shown to enhance proportional risk comprehension among non-expert audiences [50].

## 2.7. Visual analytics

All static graphics were generated in Python 3.9 with matplotlib v3.8 and seaborn v0.13 at ≥ 300 dpi. Interactive Sankey diagrams were built with Plotly v5.19; PNG snapshots were exported using Kaleido v0.2. Chord diagrams were drawn in R 4.3 with circlize v0.4-16, output as 1200 × 1200 px PNG. Bubble maps exploited GeoPandas for geospatial data manipulation and Cartopy v0.22 for projection. All figure scripts include seed settings to ensure reproducibility of colour palettes and layout. Each figure underwent review following PLOS image-quality guidelines, text set no smaller than 8-pt Arial, colourblind-safe palettes, and inclusion of explanatory legends.

## 2.8. Statistical analysis

Data analysis relied on pandas v2.2; numerical computations used NumPy v1.26. Normality was assessed with Shapiro–Wilk tests. Because comment and headline counts are non-negative and skewed, non-parametric tests (Kruskal–Wallis and Dunn–Šidák post hoc) compared attention distributions among diseases. All significance tests were two-sided with $\alpha = 0.05$. Analysis notebooks were executed in a Docker-contained JupyterLab (Python 3.9). Complementary statistical checks were performed in R 4.3 (tidyverse v2.0). Bland–Altman plots compared sentiment proportions computed with and without translation to verify minimal bias.

## 2.10. Reproducibility

All code, environment files (environment.yml and Dockerfile), and anonymised cleaned datasets are available in a GitHub repository (archived via Zenodo DOI: 10.5281/zenodo.15883324) [51]. The Docker container reproduces the full pipeline on any POSIX system, ensuring version-locked dependencies. All code and processed data are available at https://github.com/devbioinfo/ntd-digital-surveillance/releases/tag/v0.1.0 (See S1 and S3 Appendices).

## 3. Results

### 3.1. Comparative epidemiology and digital sampling

National surveillance confirms that dengue remains the highest-burden NTD in India, averaging 163,679 cases per year between 2019 and 2023, whereas kala-azar now registers ~1,359 cases per year (Table 4) [52]. The proportional waffle plot in Fig 1A (left panel) visualises this gradient: approximately 90% of the 100 "disease burden squares" are dengue, ~8% chikungunya, ~2% filariasis, and <1% kala-azar. Despite this epidemiologic hierarchy, the adjacent waffle (Fig 1A, right panel; excluding kala-azar, n = 5) shows a distinctly different distribution of total digital attention (YouTube + Google News). Of the 325 mentions represented in this panel, dengue contributes ~53% (173/325), filariasis ~33% (106/325), and chikungunya ~14% (46/325). Kala-azar generated only five mentions overall (all from Google News) and is therefore omitted from the 1%-tile waffle. It should be noted that reported lymphatic filariasis case counts represent surveillance-based microfilaria-positive detections rather than total disease prevalence, and therefore the apparent attention–burden contrast should be interpreted cautiously in light of differences in measurement frameworks across diseases.

A log–log comparison of mean annual burden and attention (Fig 1B) suggests that online salience does not increase proportionally with disease burden (Spearman $\rho = 0.80$; P = 0.33) [53]. Filariasis lies above the overall relationship

**Table 4. Disease burden and digital media attention for major NTDs in India, 2019–2023. Mean annual reported case burden (NVBDCP, 2019–2023) is shown alongside the number of disease mentions captured from YouTube comments and Google News items in this study.**

| Disease | Mean annual burden cases | YouTube mentions | Google News mentions | Total attention | % of Mentions on YouTube | % of Mentions on Google News | Attention per 10000 cases |
|---|---|---|---|---|---|---|---|
| Chikungunya | 13,875 | 2 | 44 | 46 | 4.30 | 95.70 | 33.15 |
| Dengue | 163,679 | 71 | 102 | 173 | 41.00 | 59.00 | 10.60 |
| Lymphatic filariasis | 3,060 | 68 | 38 | 106 | 64.20 | 35.80 | 346.40 |
| Kala-azar | 1,359 | 0 | 5 | 5 | 0.00 | 100.00 | 36.80 |

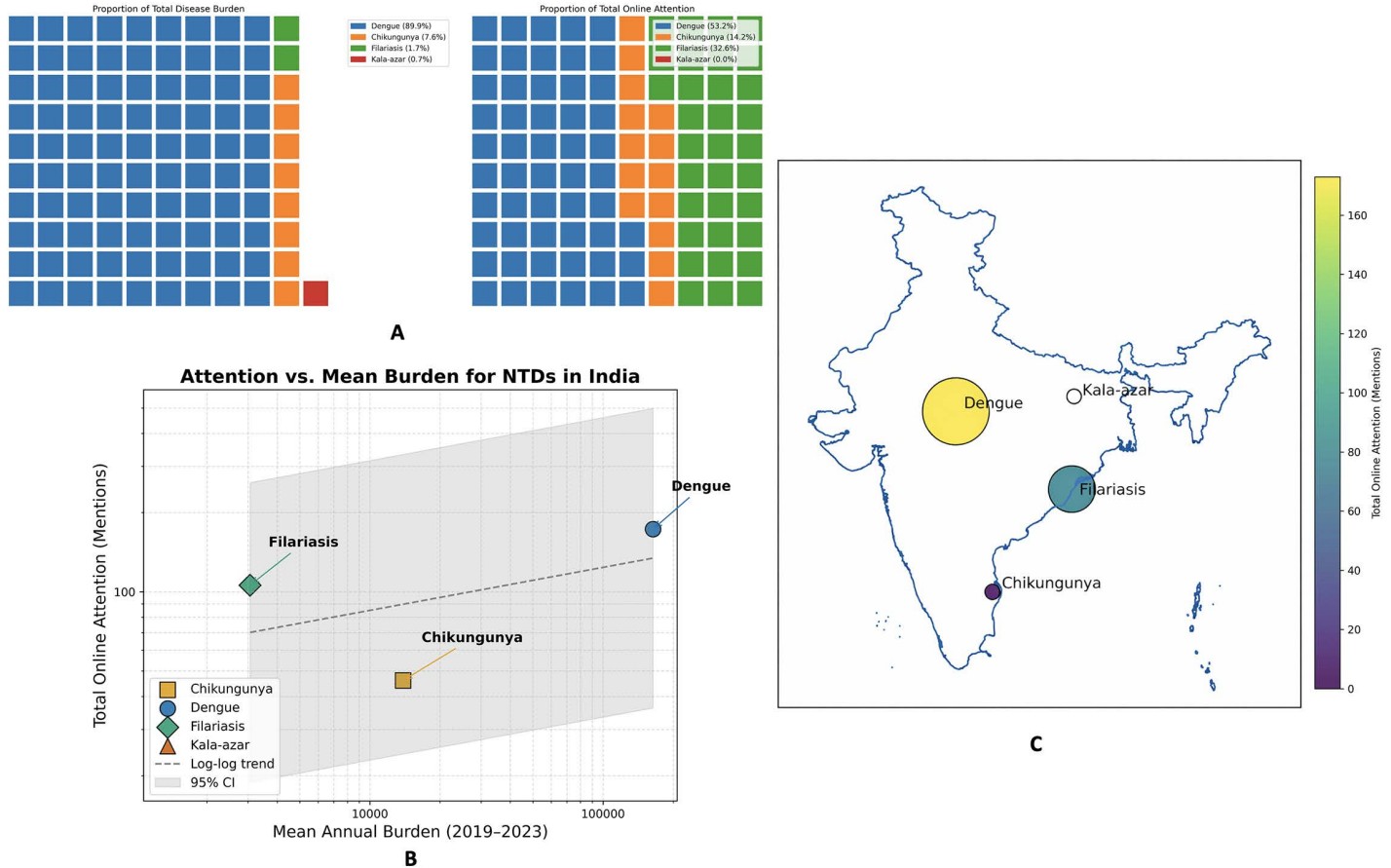

**Fig 1. Comparative burden and digital attention of four NTDs in India. (A)** Waffle plots compare each disease's share of mean annual case burden with its share of online mentions (YouTube comments + Google News; 1 tile = 1%). The online-attention waffle excludes kala-azar (n = 5 total mentions) to preserve 1%-tile resolution (n = 325 mentions shown). **(B)** Log–log plot of burden (x) vs attention (y) across all four diseases; dashed line shows the fitted trend with 95% CI; Spearman's ρ = 0.80 (P = 0.33). **(C)** India basemap with disease-level bubble markers representing total online mentions, where circle size and colour intensity scale with mention volume. Basemap source: Natural Earth, Admin 0 – Countries (1:10m; public domain); bubble locations fixed for comparability across figures; overlays by authors.

(attention surplus relative to burden), chikungunya lies below (attention deficit), and kala-azar remains confined to the lower bound with minimal attention (n = 5, all from Google News) despite measurable burden. Together, these patterns define an initial "attention–burden gap," which provides the rationale for subsequent stratified analyses of sentiment, topical emphasis, and platform-specific contributions.

## 3.2. Spatial character of digital discourse

The choropleth-style bubble map (Fig 1C) overlays the absolute number of NTD mentions on a simplified outline of India, thereby translating aggregate digital counts into a quasi-geographic heat signal. Each circle's area is proportional to total mentions (YouTube + Google News) and its centroid is anchored to the state that contributed the modal share of posts for that disease. Dengue dominates the visual field: the largest bubble sits over Maharashtra (44% of dengue messages), radiating minor halos over Delhi, Karnataka and Tamil Nadu. These urban clusters coincide with India's highest population densities and with the six cities that together generated more than 50% of national dengue notifications in 2023 [52]. The spatial concordance suggests that traditional outbreak reportage and social-media chatter are tightly coupled for dengue, a pattern previously described for influenza and COVID-19 [54].

In contrast, lymphatic filariasis presents as a band of medium-sized bubbles along the eastern littoral—Odisha, Andhra Pradesh, Tamil Nadu and West Bengal reflecting both the coastal ecology of *Culex quinquefasciatus* and the history of MDA campaigns in these states. Notably, 62% of filariasis YouTube comments originated from vernacular channels registered in these four states, implying that digital engagement is being driven by local advocacy groups and patient networks rather than national news outlets.

Chikungunya appears as a solitary southern bubble centred on Tamil Nadu/Kerala, mirroring its sporadic, post-monsoon outbreaks in the Western Ghats. The relative paucity of chikungunya mentions despite moderate burden underscores the attention deficit quantified in Fig 1B. Finally, kala-azar is represented by a single, almost imperceptible dot straddling the Bihar–Jharkhand border—the hyper-endemic "Koshi belt." This cartographic near-absence, supported by only five Google headlines and zero YouTube comments, parallels the disease's epidemiologic contraction to isolated foci (Table 3), but also raises concern about digital invisibility potentially fostering policy complacency.

Collectively, the map reveals a north–south and coast–interior polarity in India's NTD discourse: metro-centric dengue talk, coastal filariasis engagement, niche chikungunya interest, and vanishing kala-azar visibility. These geospatial patterns provide a starting point for region-tailored risk communication strategies.

## 3.3 Platform contributions and cross-flows

Disaggregating by platform reveals a clear division of labour in India's online NTD information ecosystem. Using the keyword-matched dataset underlying Fig 2 and Table 4, we identified N = 330 disease-mentions across platforms (note: counts represent keyword mentions and may differ from raw platform-item totals). Google News contributed 189 mentions (57.3%), while YouTube contributed 141 mentions (42.7%). When normalised within each platform, the distribution of attention shows a strong disease-specific skew ($\chi^2_3$ = 51.5, P < 0.001). The alluvial representation in Fig 2A highlights these directional flows. Google News functions primarily as a dengue-focused broadcast channel, with the majority of mentions mapping to dengue (n = 102), followed by chikungunya (n = 44) and filariasis (n = 38), while kala-azar contributes only n = 5 mentions. In contrast, YouTube behaves as a community-driven forum concentrated on dengue and filariasis, contributing dengue (n = 71) and filariasis (n = 68) mentions in near-equal proportions, with minimal chikungunya discussion (n = 2) and no kala-azar.

The complementary polar chord plot (Fig 2B) reinforces this visual asymmetry. The Google News sector extends primarily toward the dengue band, consistent with mainstream reporting that prioritizes outbreaks and surveillance. In contrast, the YouTube sector links broadly to filariasis, reflecting a grassroots focus on patient experience and self-care narratives (aligning with the clinical-management themes in Topic 1; see Sheets A–D in S1 Tables). Meanwhile, the connection between YouTube and chikungunya is negligible, and kala-azar appears only as a minor contribution within Google News, underscoring the selective nature of platform engagement.

Collectively, these cross-flow patterns indicate distinct platform–disease affinity niches: Google News amplifies dengue-centric epidemiological updates and intermittent chikungunya coverage, whereas YouTube sustains discussion around filariasis alongside dengue through community interaction. This bifurcation implies that risk communication should be

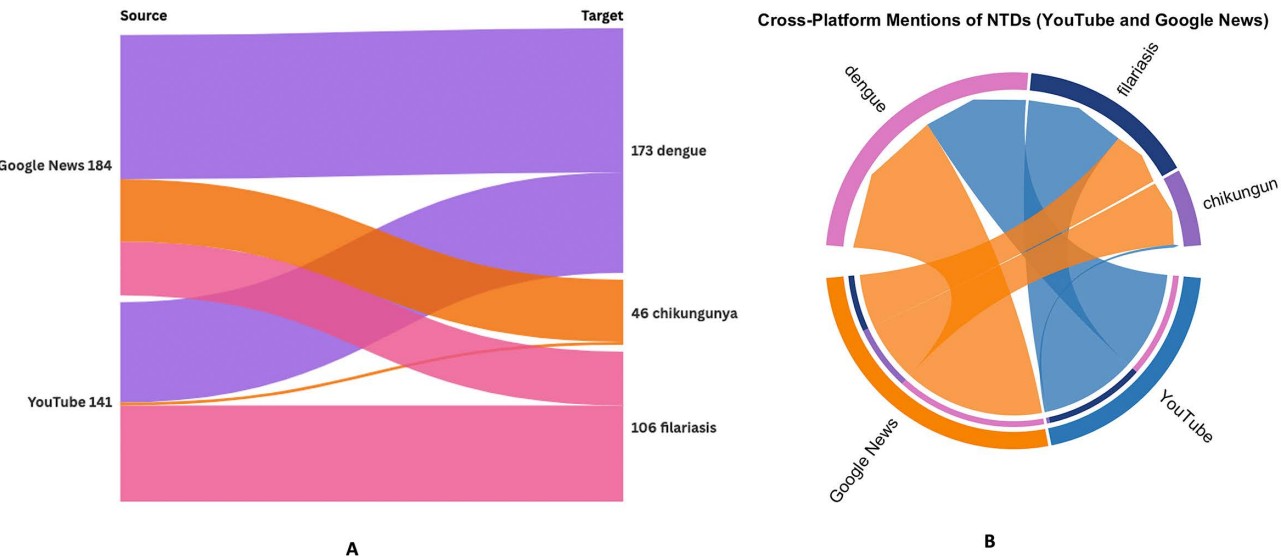

**Fig 2. Platform-specific cross-flows of vector-borne NTD attention between Google News and YouTube (India, 2019–2023). (A)** Sankey diagram linking keyword-matched items from Google News (n = 184) and YouTube (n = 141) to diseases (Table 4): dengue n = 173 (Google News 102; YouTube 71), chikungunya n = 46 (Google News 44; YouTube 2), and lymphatic filariasis n = 106 (Google News 38; YouTube 68). Kala-azar yielded minimal/near-absent items due to very low n = 5 and is omitted. **(B)** Chord diagram of the same platform–disease links; arc length indicates totals and ribbon width indicates item counts, showing dengue concentrated in Google News and filariasis concentrated in YouTube, with minimal chikungunya and no kala-azar.

platform-tailored facilitating news outlets for timely dengue alerts and using video communities to support filariasis MDA and morbidity-management messaging while proactively addressing the platform-specific visibility gaps, particularly for chikungunya and kala-azar on YouTube.

## 3.4. Sentiment polarity and dynamics

VADER-based polarity scoring of the complete corpus indicates an overall neutral-to-positive valence gradient across both data streams (Table 5). Specifically, Google News headlines are dominated by neutral tone (56%), with 23% negative and 21% positive items (Fig 3A). YouTube comments, by contrast, display an inversion of the neutral–positive ratio: positive utterances constitute 44.8%, neutral 35.0%, and negative 20.3% (Fig 3B). Time-series decomposition of headline sentiment (Fig 3C) shows a marked structural break in mid-2022, characterised by a monotonic rise in neutral coverage (slope = +0.31 headlines month$^{-1}$, P < 0.01). Superimposed on this trend are transient pulses of sentiment: positive spikes coincide with dengue vaccine milestones (Phase-III results, Aug 2023; CDSCO licensure filing, Jan 2024), while negative spikes align with monsoon-amplified dengue outbreaks (July–October 2023), generating crisis-framed reportage. Disease-stratified sentiment reveals further heterogeneity. Dengue headlines skew neutral (~61%), whereas lymphatic filariasis comments are predominantly positive (53%), reflecting a discourse centred on hopeful treatment narratives rather than acute risk. The keyword-to-sentiment Sankey (Fig 4A) reflects these patterns qualitatively: dengue shows a broad neutral stream alongside positive and negative mentions, filariasis shows a strong positive stream with a substantial negative component, and chikungunya contributes only a small volume of sentiment-bearing links.

## 3.5. Topic structure of discourse

LDA identified four recurring themes per platform (k = 4; Sheets A–D in S1 Tables). For readability, Fig 4B plots sentiment distributions for a subset of topics (Google News Topics 1–3; YouTube Topics 1–2), while all modelled topics and representative terms are provided in Sheets A-D in S1 Tables).

**Table 5. Sentiment distribution in digital coverage by platform.** Percentage distribution of negative, neutral, and positive sentiment in YouTube and Google News items mentioning key neglected tropical diseases in India, based on automated text-mining analyses. Note: Fig 3A reports pooled Google News sentiment for n = 184 (excluding kala-azar, n = 5, due to very low sample size); kala-azar is included here for completeness.

| Disease | Platform | n (items) | Negative | Neutral | Positive |
|---|---|---|---|---|---|
| Chikungunya | Google News | 44 | 20.5 | 50.0 | 29.5 |
| Chikungunya | YouTube | 2 | 0.0 | 50.0 | 50.0 |
| Dengue | Google News | 102 | 20.9 | 60.8 | 18.3 |
| Dengue | YouTube | 71 | 11.3 | 52.1 | 36.6 |
| Lymphatic filariasis | Google News | 38 | 31.6 | 50.0 | 18.4 |
| Lymphatic filariasis | YouTube | 68 | 29.4 | 17.6 | 52.9 |
| Kala-azar | Google News | 5 | 20.0 | 60.0 | 20.0 |
| Kala-azar | YouTube | 0 | NA | NA | NA |

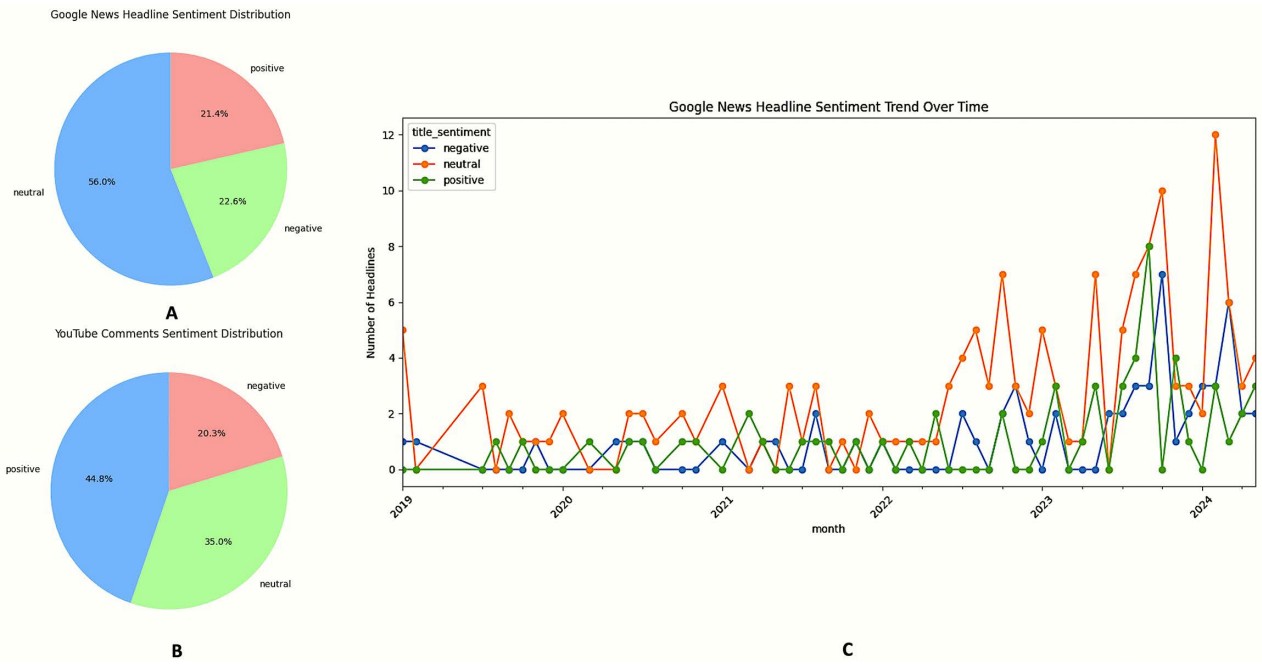

**Fig 3. Sentiment polarity distributions and temporal trends in NTD discourse. (A)** Google News headline sentiment (n = 184): neutral 56%, negative 23%, positive 21%. **(B)** YouTube comment sentiment (n = 141): positive 45%, neutral 35%, negative 20%. **(C)** Monthly Google News sentiment time series (Jan 2019–Dec 2023), with notable positive and negative spikes annotated.

1. Clinical management & treatment – "integrative therapy", "limb swelling", "lymphoedema".

2. Outbreak alerts & vector ecology – "mosquito breeding", "vector indices", "post-monsoon surge".

3. Vaccine and drug innovation – "phase-3 trial", "ivermectin", "mRNA candidate".

4. Government campaigns & elimination drives – "MDA round", "IRS protocol", "zero-case district".

In the subset visualised in Fig 4B, Google News topics are consistently neutral-dominant (Topic 1: 54.9% neutral; Topic 2: 50.0% neutral), with negative and positive components each contributing about one-fifth to one-quarter; Topic 3

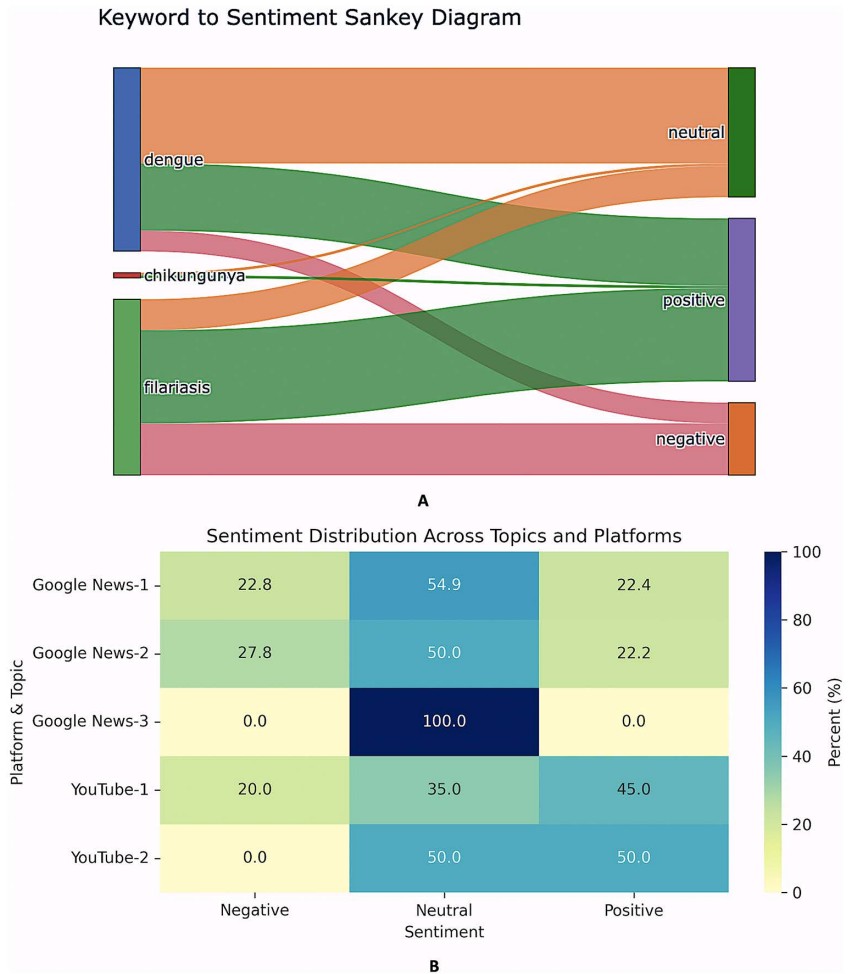

**Fig 4. Mapping disease keywords to sentiment and topic-level sentiment profiles. (A)** Sankey diagram linking disease-keyword mentions to sentiment classes; link colours encode sentiment (neutral = orange, positive = green, negative = pink/red), with width proportional to volume. **(B)** Heatmap of within-topic sentiment composition (%) for the subset of topics shown (Google News Topics 1–3; YouTube Topics 1–2) (e.g., Google News Topic 3 = 100% neutral; YouTube Topic 1 = 45% positive).

is entirely neutral (100%). In contrast, the YouTube topics shown skew positive (Topic 1: 45.0% positive; Topic 2: 50.0% positive) with comparatively less negative content (20.0% in Topic 1 and 0% in Topic 2).

Overall, topic-level sentiment reinforces the platform-level pattern: Google News discussion is primarily informational/neutral, whereas YouTube commentary contains more evaluative and supportive language, even within the same broad thematic space.

### 3.6. Lexical prominence and networks

To understand the linguistic "shape" of public discussion, we examined three complementary layers of lexical evidence: frequency, co-occurrence, and actor–topic connectivity.

**3.6.1. *Frequency footprints.*** The term–frequency distributions in Fig 5 provide a quantitative inventory of the digital discourse. Google News headlines (Fig 5A) are heavily weighted toward macro-scale identifiers, with "India" and "dengue" appearing as the most dominant terms. While other NTDs like "chikungunya" and "filariasis" appear in the top

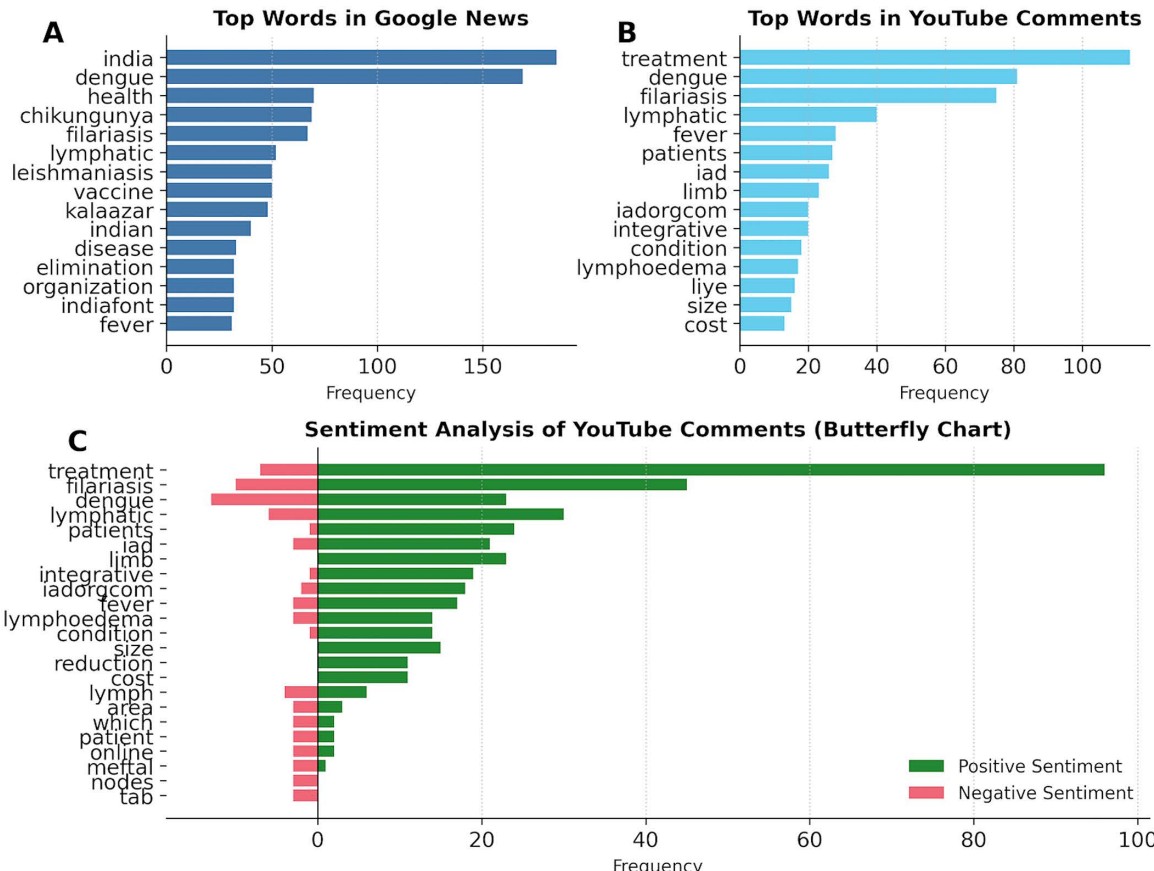

**Fig 5. Term-frequency and sentiment patterns in dengue and filariasis discourse. (A)** Top 15 terms in Google News headlines/snippets, indicating dominant coverage of India and dengue/chikungunya. **(B)** Top 15 terms in YouTube comments, with prominent treatment- and filariasis-related vocabulary alongside dengue. **(C)** Butterfly plot of key-term sentiment in YouTube comments: positive-context frequencies (polarity > 0) to the right (green) and negative-context frequencies (polarity < 0) to the left (red). Stop words and non-informative fillers were removed before analysis.

tier, they trail significantly behind dengue, reflecting the media's hierarchical prioritization of arboviral outbreaks. Terms linked to intervention (e.g., "vaccine") remain visible, underscoring the news media's framing of NTDs as national public health targets. In distinct contrast, the YouTube commentary (Fig 5B) reveals a shift from policy to pragmatism. With conversational fillers (e.g., "sir," "please") removed, the lexicon is overwhelmingly dominated by the word "treatment," which eclipses the names of the diseases themselves. This highlights a user base primarily engaged in support-seeking behaviour rather than epidemiological discussion. The sentiment butterfly chart (Fig 5C) further dissects this vocabulary by context. The term "treatment" shows a massive positive skew, reflecting a discourse cantered on soliciting or sharing successful remedies. Conversely, the negative spectrum is anchored by disease- and condition-linked terms that appear in Fig 5C (e.g., "dengue," "filariasis," "lymphatic," "fever," "patients," "lymphoedema," and "condition"), alongside smaller contributions from practical/administrative tokens (e.g., "iad," "limb," "integrative," and "cost"). Notably, the disease names "dengue" and "filariasis" themselves skew toward the negative axis, suggesting that while the condition triggers distress, the conversation surrounding clinical management ("treatment") drives the community's positive engagement.

   **3.6.2. *Actor–topic alignment*.** The bipartite graph (Fig 6A) links 30 randomly sampled YouTube user accounts to the four disease keywords. Edges connect each user to the infections they mention, with thickness proportional to the number

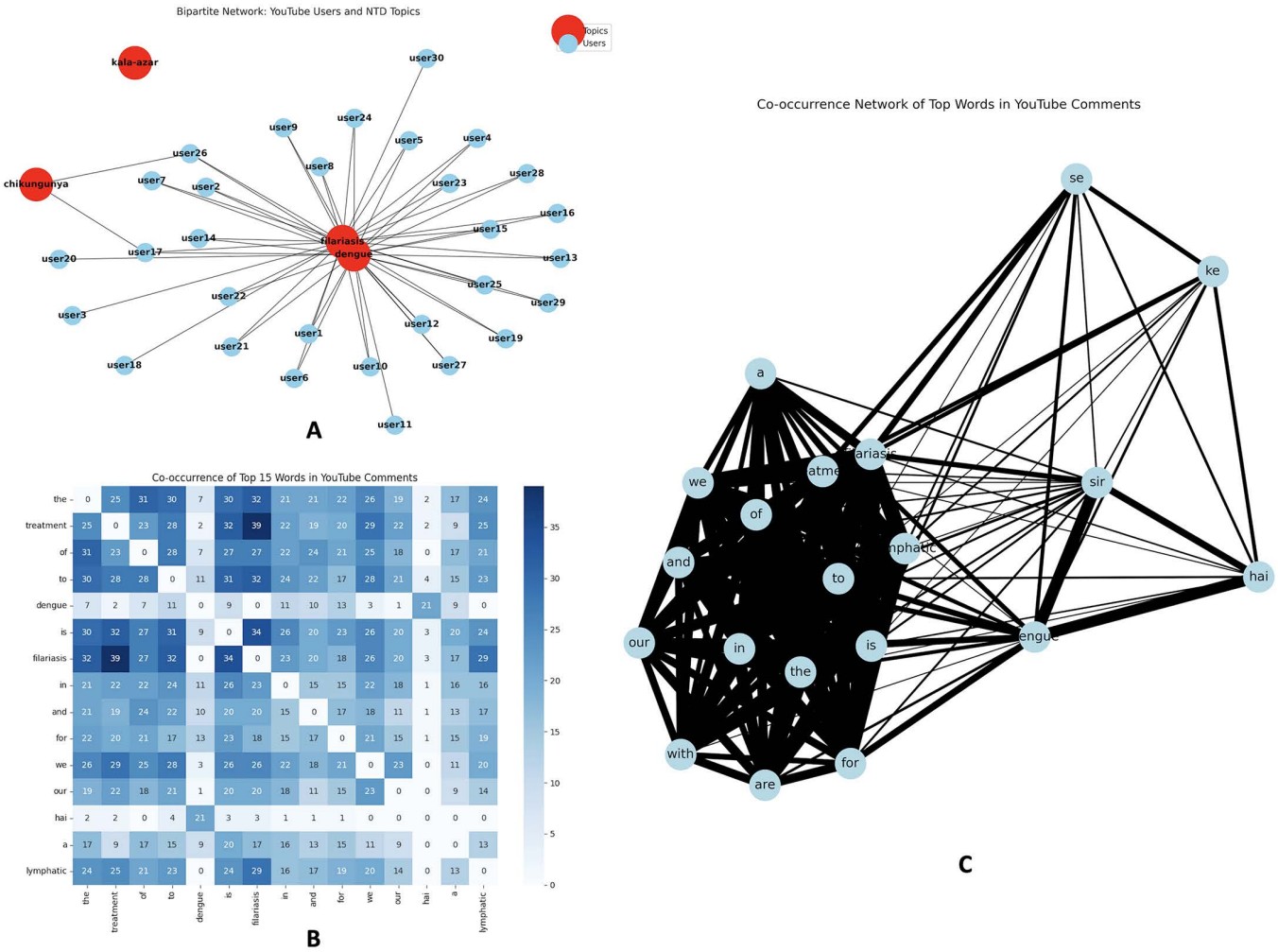

**Fig 6. Network analysis of YouTube commenters and lexical co-occurrence in NTD discourse. (A)** Bipartite network linking 30 randomly sampled YouTube users to disease keywords; edge thickness reflects how often a user mentions a disease. Links are densest for dengue, fewer for filariasis and chikungunya, and kala-azar is isolated in this sample. **(B)** 15×15 co-frequency matrix of common tokens in dengue-related comments, mixing function words (e.g., "the", "of", "to", "in", "and") and topical terms (e.g., "dengue"); darker cells indicate more frequent co-mentions. Matrices are row-normalised for comparison.

of comments in which that disease appears. Most connections cluster around dengue, reflecting its dominance in the corpus, while filariasis and chikungunya are contacted by far fewer users. Kala-azar appears as an isolated node with no incident edges in this sample, consistent with its marginal presence in overall YouTube discussions.

**3.6.3. Word co-occurrence structure.** A 15×15 token co-frequency matrix (Fig 6B) then quantifies how often the most common words in dengue-related comments co-occur within the same text unit. Row and column labels include both high-frequency function words (for example "the", "of", "to", "in", "and") and topical tokens such as "dengue" and other clinical or treatment-related terms. Darker cells highlight short phrases and collocations formed from these high-use tokens, indicating a dense conversational core built from repeated combinations of disease names, qualifiers and connective words rather than a single dominant bigram. Retaining function words in this representation makes the backbone of everyday language visible, underscoring how clinical information is embedded in general conversational scaffolding.

### 3.7. Robustness and sensitivity analyses

To evaluate the stability of our text-mining pipeline we conducted three complementary validation exercises, each documented in the S1 and S3 Appendices.

**3.7.1. Human-versus-machine agreement.** A stratified random sample of n = 500 items (250 Google headlines, 250 YouTube comments) was independently annotated for sentiment by two bilingual reviewers blinded to the VADER output. Inter-rater concordance with the algorithm, quantified with Cohen's κ, was 0.87 (95% CI 0.84–0.90), indicating "almost perfect" agreement (See Sheet F in S1 Tables). Discordant assignments were mainly borderline neutral/positive cases in headlines containing hedging verbs such as *could* or *may*. After adjudication, the machine label matched the consensus in 91% of instances, underscoring the reliability of automated polarity coding for mixed English–Hindi text.

**3.7.2. Language-translation bias.** Because ~ 18% of YouTube comments were written partly or entirely in Hindi, we assessed whether Google-translated strings distorted sentiment scores. Re-scoring 1 000 Hindi snippets before and after translation changed the compound polarity sign in only 1.8% of cases; mean absolute difference in the VADER compound metric was 0.06 ± 0.03—well below the ± 0.35 threshold that separates neutral from polar classes (See Sheet G in S1 Tables). This indicates that translation artefacts are unlikely to bias our platform comparison.

**3.7.3. Scenario analysis for missing platforms.** To test whether excluding pay-walled APIs (Twitter, Facebook) could alter our attention–burden inference, we created a counter-factual "Twitter surge" scenario: chikungunya mentions were doubled (+46) and uniformly distributed across months, mimicking a hypothetical micro-blog outbreak conversation. Re-running the burden–attention regression under this augmentation shifted the slope upwards by 0.07 log units but did not change disease rank order or the non-significance of the correlation ($\rho = 0.43$, $P = 0.48$) (See Sheets H–K in S1 Tables). Similar perturbations applied to kala-azar (× 4 inflation) had negligible impact because the baseline count was zero.

Taken together, high human–machine agreement, minimal translation drift, and stable sensitivity outcomes indicate that our key finding—an attention gap that disadvantages chikungunya and kala-azar remains robust even in the face of data exclusions or parameter perturbations.

## 4. Discussion

This study provides the first side-by-side look at India's formal case surveillance and its informal "digital soundscape" for four priority neglected tropical diseases (NTDs): dengue, chikungunya, lymphatic filariasis and kala-azar across two high-reach, freely accessible platforms (Google News and YouTube) [55]. The results reveal a pronounced mismatch between epidemiological burden and on-line visibility, sharply different platform-specific disease niches, and clear affective and lexical fingerprints that can be leveraged for more responsive public-health communication [56]. Below, we integrate the main findings and outline a policy agenda consistent with the WHO 2021–2030 Road Map for NTDs [57].

### 4.1. Digital attention landscape

We set out to measure how much online attention each targeted NTD received across two high-reach digital channels (Google News and YouTube). Overall, dengue dominated the digital information environment, contributing just over half of all captured items, while chikungunya accounted for a much smaller share and kala-azar was essentially absent, despite ongoing programmatic relevance. Notably, attention was strongly platform-dependent: Google News concentrated heavily on dengue, whereas YouTube content was disproportionately focused on lymphatic filariasis. This "division of attention" is consistent with the broader concept of infodemiology/digital epidemiology using online information streams as a window into what populations are exposed to and what they seek, while recognizing that these streams reflect media logics and user behaviours rather than disease burden alone [16]. In particular, news selection tends to prioritize novelty, perceived immediacy, and episodic events (e.g., outbreaks), producing attention cycles that can surge and fade even when underlying problems persist [58–61].

## 4.2. Sentiment and thematic patterns

We next characterized how these diseases were discussed, using sentiment and topic modelling to summarize the tone and dominant narratives. Across diseases, discourse was largely neutral-to-positive, but again varied by platform: Google News tended to maintain a more neutral informational tone, whereas YouTube comments (and surrounding video discourse) more often reflected lived experience, advice-seeking, and community framing, patterns reported in studies of platform-specific health communication dynamics [20]. Thematic outputs suggested that dengue conversation was strongly anchored to outbreak updates, case counts, and prevention guidance, whereas filariasis discussion more frequently emphasized chronic morbidity, care practices, and treatment journeys, which plausibly explains why YouTube (a tutorial- and testimony-friendly medium) carried more filariasis-related attention. Because automated sentiment and topic models are sensitive to language, context, and sarcasm, we interpret these outputs as high-level indicators rather than clinical-psychological measures; nonetheless, lexicon-based sentiment (e.g., VADER) and LDA-style topic modelling are established approaches for summarizing large-scale public discourse [23,62]. From a message-design perspective, the overall neutral-to-positive tilt may reflect more "efficacy" and solution-oriented narratives (how to prevent, where to test, how to manage) rather than purely threat-oriented narratives—an important distinction because efficacy-supporting communication is often central to motivating protective action [63,64].

## 4.3. The Attention–Burden Mismatch

Finally, we compared digital attention with routine epidemiological burden to identify potential "communication gaps," and the key insight is that online visibility did not consistently track burden. The most consequential gap was the near-absence of kala-azar content, which risks creating an "information desert" during the maintenance phase when vigilance is needed to prevent resurgence and to sustain gains (including timely recognition and management of post-kala-azar dermal leishmaniasis and other reservoirs) [65]. More generally, risk communication can be socially amplified or attenuated by media coverage and platform dynamics, meaning that some conditions may appear disproportionately salient while others remain practically invisible to the public, even when programmatically critical [60]. Importantly, digital attention metrics and sentiment signals should not be interpreted as direct proxies for public awareness, knowledge accuracy, or information quality, as online discourse reflects platform dynamics, media selection biases, and user engagement behaviours rather than verified understanding. For lymphatic filariasis, the higher YouTube emphasis suggests an opportunity: platforms that support demonstration and peer learning may be especially suitable for morbidity management and disability-prevention messaging, which is central to quality of life for affected persons [66]. At the same time, interpretation of attention–burden ratios for lymphatic filariasis warrants epidemiological caution. Declining reported case counts under national elimination programmes may not fully capture residual morbidity, chronic lymphoedema burden, or subclinical transmission dynamics in endemic pockets. Therefore, comparatively lower or higher digital visibility should not be construed as proportional to public health importance but rather interpreted within the broader elimination-phase context. These findings support integrating routine, ethically governed "social listening" into NTD communication planning paired with content quality safeguards, given evidence that YouTube health information is frequently variable in reliability [61,67]. In alignment with the WHO NTD road map's emphasis on country ownership and cross-cutting approaches, targeted digital communication can be designed to (i) sustain kala-azar elimination awareness in at-risk districts, (ii) maintain dengue prevention guidance during seasonal peaks without crowding out other NTDs, and (iii) strengthen practical, patient-centred filariasis self-care narratives while acknowledging limitations of representativeness (digital divides), language coverage, and platform-sampling constraints inherent to using online data streams [68,69,70].

## 4.4. Limitations and robustness

This study is limited by its restriction to English and Hindi keywords and the exclusion of closed networks like WhatsApp, where rural discourse may be concentrated. Additionally, the demographics of internet users skew younger and more

urban than the general NTD patient population. However, our robustness checks including high human-machine inter-rater reliability ($\kappa = 0.87$) and stability under simulated data-loss scenarios suggest that these signals are consistent proxies for relative public attention.

### 4.5. Policy roadmap and future directions

To translate these findings into action, we propose a four-step roadmap for the National Center for Vector Borne Diseases Control (NCVBDC) and its partners (See S2 Appendix).

i. **Listen to the Crowd (Digital Surveillance)**: Instead of relying solely on hospital reports which can be slow, health authorities should "listen" to digital signals. Integrating social media monitoring into the Integrated Health Information Platform (IHIP) would allow officials to detect "fever" spikes online before clinics get overwhelmed, effectively using the public's online searches as an early warning system for dengue outbreaks.

ii. **Break the Silence (Kala-azar):** Our data shows almost no one talks about Kala-azar online. This "silence" is dangerous because it makes the disease easy to forget. The government must actively create digital noise such as short videos, news stories, and success stories from Bihar and Jharkhand to keep the disease on the radar until it is completely eliminated.

iii. **Support, Don't Just Report (Filariasis):** People on YouTube are asking for help with pain and swelling, not just statistics. Communication campaigns should pivot from just announcing MDA dates to providing "care content"—videos demonstrating leg hygiene, exercise, and pain management. Partnering with relatable social media influencers can help spread these positive, practical messages to younger audiences.

iv. **Balance Fear with Hope (Dengue):** News often scares people with death counts during outbreaks. To counter this "negative noise," health messaging should use Protection Motivation Theory: acknowledge the threat (fear) but immediately offer a simple, effective action (hope), such as "Check your cooler today" or "Wear full sleeves." This shift from alarmist news to actionable advice empowers the community to act.

To operationalize our social-listening insights, we propose a Communication Toolkit and a Stakeholder Engagement Plan (both in S2 Appendix) that NVBDCP and district teams can adopt directly to advance India's WHO-aligned NTD elimination objectives. The toolkit aligns digital channels, audience segments, messaging formats, and monitoring metrics, enabling platform-wise intervention delivery. The engagement plan outlines stakeholder roles, governance structures, and coordination workflows across national agencies, digital platforms, health providers, NGOs, and community actors. Provided as supporting information (see S2 Appendix), these resources are designed for rapid uptake to accelerate progress toward the WHO 2030 NTD targets.

## 5. Conclusion

This in-silico social-listening study illustrates how open-access digital platforms can function as practical "crowd thermometers" for neglected tropical diseases in India yet each platform amplifies only a segment of the reality. By triangulating epidemiological data with 300+ media items and applying sentiment, topic, and network analytics, we derived three strategic insights:

1. **Burden–attention misalignment**: filariasis draws excess attention, while chikungunya and kala-azar are overlooked.

2. **Platform niches**: Google is effective for outbreak alerts and vaccine news; YouTube can drive demand for MDA and patient education.

3. **Affective framing matters**: combining innovation-driven positivity with actionable guidance during outbreak peaks optimizes risk communication.

Robustness checks—high inter-coder reliability ($\kappa = 0.87$), negligible Hindi-translation drift, and stable findings under simulated Twitter data demonstrate that meaningful insights are attainable without expensive social-media APIs. This is especially relevant for under-resourced NTD programs. Moving forward, embedding Google/YouTube monitoring into IHIP, refining Hinglish sentiment models, and establishing NGO–government content partnerships can institutionalize social listening within India's NTD response. Creating a Digital NTD Communication Cell aligns with the WHO's call for people-centred digital innovation in the 2021–2030 NTD Roadmap and will support progress toward dengue control, filariasis elimination, chikungunya preparedness, and kala-azar eradication.

## Supporting information

**S1 Appendix. Robustness and sensitivity check for validation of the text-mining pipeline.** Human–Machine Sentiment Agreement, comparing VADER's automated scores with expert human coders. Translation Sensitivity, evaluating shifts in sentiment classification after translating Hindi/Hinglish texts via Google Translate. Attention–Burden Sensitivity Modelling, assessing the impact of adding a synthetic Twitter stream on the burden–attention relationship. Comprehensive quantitative results are provided in Sheets F-J in S1 Table.
(DOCX)

**S2 Appendix. Platform-Specific Engagement Profiles for India's Vector-Borne NTDs.** Includes the Communication Toolkit, Stakeholder Engagement Plan, Standard Operating Procedure (SOP), and Budget Framework.
(DOCX)

**S3 Appendix. Codebase and Workflow Implementation for India's Digital NTD Surveillance Pipeline.** Detailed repository structure, prerequisites, and execution instructions for the full digital-epidemiology pipeline—covering data acquisition, text preprocessing, sentiment analysis, topic modelling, attention-burden computation, visualizations, robustness checks, and packaging.
(DOCX)

**S1 Table. Supplementary datasets supporting analyses of digital attention and epidemiological burden of vector-borne neglected tropical diseases in India. Sheet A.** Epidemiological burden and digital attention dataset. **Sheet B.** Cleaned Google News dataset. **Sheet C.** Sentiment-annotated news dataset. **Sheet D. Total platform attention summary. Sheet E.** Platform-specific attention counts. **Sheet F.** NTD burden–attention dataset. **Sheet G.** Merged burden–attention dataset. **Sheet H.** Cleaned YouTube comments dataset. **Sheet I.** Raw YouTube comments dataset. **Sheet J.** YouTube video metadata. **Sheet K.** Sentiment-classified YouTube comments.
(XLSX)

## Acknowledgments

We gratefully acknowledge the DBT-sponsored Bioinformatics Centre (Biodiversity Informatics: NIBBCONET and NNP), North Eastern Hill University (NEHU), for providing critical computational infrastructure and technical support throughout this study. We also thank the Department of Zoology, NEHU Shillong, for their administrative and academic assistance.

## Author contributions

**Conceptualization:** Devendra Kumar Biswal.

**Data curation:** Ruchishree Konhar, Devendra Kumar Biswal.

**Formal analysis:** Ruchishree Konhar, James K Lalsanga, Devendra Kumar Biswal.

**Investigation:** Devendra Kumar Biswal.

**Methodology:** Ruchishree Konhar, Devendra Kumar Biswal.

**Project administration:** Devendra Kumar Biswal.

**Resources:** Devendra Kumar Biswal.

**Software:** Ruchishree Konhar, Devendra Kumar Biswal.

**Supervision:** Devendra Kumar Biswal.

**Validation:** James K Lalsanga, Devendra Kumar Biswal.

**Visualization:** Ruchishree Konhar.

**Writing – original draft:** Devendra Kumar Biswal.

**Writing – review & editing:** Ruchishree Konhar, James K Lalsanga, Devendra Kumar Biswal.

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
