## [Decision Letter · Decision Letter 0]

22 Sep 2025

Noise and neglect: Social-media signals expose attention gaps for dengue, chikungunya, lymphatic filariasis and kala-azar in India’s vector-borne NTDs

Dear Dr. Biswal,

Thank you for submitting your manuscript to PLOS Neglected Tropical Diseases. After careful consideration, we feel that it has merit but does not fully meet PLOS Neglected Tropical Diseases's publication criteria as it currently stands. Therefore, we invite you to submit a revised version of the manuscript that addresses the points raised during the review process. In particular, several reviewers pointed to the need to provide better quality and simpler and clearer figures.

Please submit your revised manuscript within 60 days Nov 21 2025 11:59PM. If you will need more time than this to complete your revisions, please reply to this message or contact the journal office at plosntds@plos.org. Please include the following items when submitting your revised manuscript:

We look forward to receiving your revised manuscript.

Kind regards,

Shih Keng Loong

Academic Editor

Shaden Kamhawi

co-Editor-in-Chief

Paul Brindley

co-Editor-in-Chief

**Additional Editor Comments (if provided):**

Reviewer #1: Authors should relook at the way the results were presented. Consider simplifying Figure 5. Also, please improve all figure quality.

Reviewer #2: Tables and figures quality must be improved.

Reviewer #3: The manuscript should be shortened to reduce redundancy and repetition, with a stronger focus on presenting only essential figures and tables. The discussion of findings should be more closely aligned with the sequence of objectives and results to improve logical flow and reader comprehension. Improve figure and table quality.

**Journal Requirements:**

**Reviewers' Comments:**

Reviewer's Responses to Questions

**Key Review Criteria Required for Acceptance?**

**Methods**

-Are the objectives of the study clearly articulated with a clear testable hypothesis stated?

-Is the study design appropriate to address the stated objectives?

-Is the population clearly described and appropriate for the hypothesis being tested?

-Is the sample size sufficient to ensure adequate power to address the hypothesis being tested?

-Were correct statistical analysis used to support conclusions?

-Are there concerns about ethical or regulatory requirements being met?

Reviewer #1: -Are the objectives of the study clearly articulated with a clear testable hypothesis stated? Yes

-Is the study design appropriate to address the stated objectives? Yes

-Is the population clearly described and appropriate for the hypothesis being tested? Yes

-Is the sample size sufficient to ensure adequate power to address the hypothesis being tested? Yes

-Were correct statistical analysis used to support conclusions? No comment

-Are there concerns about ethical or regulatory requirements being met? Yes

Reviewer #2: Appropiate but need some improvements.

Reviewer #3: The authors of submitted manuscript designed their study to extract data from social media and news media platform such as Google, tweeter and youtube about NTDS to evaluate the the public respond and pattern and impact of those on public. The study objectives are very clear and the author designed the study appropriate to perform the study. The author perform a proper data analysis and the study sound ethical.

**Results**

-Does the analysis presented match the analysis plan?

-Are the results clearly and completely presented?

-Are the figures (Tables, Images) of sufficient quality for clarity?

Reviewer #1: -Does the analysis presented match the analysis plan? Yes

-Are the results clearly and completely presented? Yes

-Are the figures (Tables, Images) of sufficient quality for clarity? Too much, with redundancy

Reviewer #2: The results are quite clear, but they should be presented in a simpler and more accessible way.

Please reconsider Figure 5 — simplify it and clarify how the authors are using this presentation.

Impove all figures quality (resolution).

Reviewer #3: All the data analyzed well using proper statistical software and method. The only issue the quality of table and figures are not clear and the authors need to improve the figure and table. The numbers and the words in figures are blur and difficult to read

**Conclusions**

-Are the conclusions supported by the data presented?

-Are the limitations of analysis clearly described?

-Do the authors discuss how these data can be helpful to advance our understanding of the topic under study?

-Is public health relevance addressed?

Reviewer #1: -Are the conclusions supported by the data presented? Yes

-Are the limitations of analysis clearly described? Yes

-Do the authors discuss how these data can be helpful to advance our understanding of the topic under study? Yes

-Is public health relevance addressed? Yes

Reviewer #2: Clear

Reviewer #3: The authors concluded their data finding very well by mentioning what are their study data can be useful and where need to be improve in the future study

**Summary and General Comments**

Reviewer #1: This manuscript presents a comprehensive and timely study on public perspectives regarding neglected tropical diseases (NTDs) using digital epidemiology, specifically through social media platforms. The approach is relevant and innovative, capturing real-world public sentiment that is often absent from formal reports, which can be limited in scope and transparency.

The strengths of the study lie in its novel use of digital platforms and its potential to inform public health messaging. However, the manuscript’s impact is reduced by its length and redundancy, which may obscure its key messages. I strongly encourage the authors to make the writing more concise and impactful.

Given that the study is based on lay public contributions, the results should also be presented in a manner accessible to non-expert audiences. Simplifying the narrative without compromising scientific rigor would increase the reach and applicability of the findings.

Reviewer #2: Figures revisions.

Reviewer #3: Overall the data presented in submitted manuscript are interesting and describe and discuss very well. The data will help with future outbreak communication and publics awareness. The author need to improve their figures and table quality before this manuscript being consider for publication.

**Figure resubmission:**
---

## [Decision Letter · Decision Letter 1]

9 Feb 2026

Response to Reviewers
Revised Manuscript with Track Changes
Manuscript

Shaden Kamhawi

co-Editor-in-Chief

Paul Brindley

co-Editor-in-Chief

**Additional Editor Comments (if provided):**
**Journal Requirements:**

**Reviewers' comments:**

**Key Review Criteria Required for Acceptance?**

**Methods**

-Are the objectives of the study clearly articulated with a clear testable hypothesis stated?

-Is the study design appropriate to address the stated objectives?

-Is the population clearly described and appropriate for the hypothesis being tested?

-Is the sample size sufficient to ensure adequate power to address the hypothesis being tested?

-Were correct statistical analysis used to support conclusions?

-Are there concerns about ethical or regulatory requirements being met?

Reviewer #1: -Are the objectives of the study clearly articulated with a clear testable hypothesis stated? Yes

-Is the study design appropriate to address the stated objectives? Yes

-Is the population clearly described and appropriate for the hypothesis being tested? Yes

-Is the sample size sufficient to ensure adequate power to address the hypothesis being tested? Yes

-Were correct statistical analysis used to support conclusions? No comment

-Are there concerns about ethical or regulatory requirements being met? Yes

Reviewer #2: Yes

Reviewer #3: The methods written well with all details mentioned

**Results**

-Does the analysis presented match the analysis plan?

-Are the results clearly and completely presented?

-Are the figures (Tables, Images) of sufficient quality for clarity?

Reviewer #1: -Does the analysis presented match the analysis plan? Yes

-Are the results clearly and completely presented? Yes

-Are the figures (Tables, Images) of sufficient quality for clarity? Trimmed & improvised.

Reviewer #2: Yes

Reviewer #3: The results look fine and the authors did proper analysis but the figures still not in the best quality. The figures are very blur and difficult to read the numbers

**Conclusions**

-Are the conclusions supported by the data presented?

-Are the limitations of analysis clearly described?

-Do the authors discuss how these data can be helpful to advance our understanding of the topic under study?

-Is public health relevance addressed?

Reviewer #1: -Are the conclusions supported by the data presented? Yes

-Are the limitations of analysis clearly described? Yes

-Do the authors discuss how these data can be helpful to advance our understanding of the topic under study? Yes

-Is public health relevance addressed? Yes

Reviewer #2: Yes

Reviewer #3: The conclusion written well and supported by data presented

**Editorial and Data Presentation Modifications?**

Reviewer #1: Authors made all neccessary corrections based on the comments.

Reviewer #2: Minor revision

Reviewer #3: (No Response)

**Summary and General Comments**

Reviewer #1: This study bridges public perceptions with public health priorities, and a press release could amplify its impact by

raising awareness of both NTDs and the role of digital surveillance in improving health communication.

Reviewer #2: The revised manuscript is clear, well structured, and significantly strengthened in its conceptual framing, methods, and interpretation. The platform-specific analysis of Google News versus YouTube is particularly strong and adds important insight into NTD communication dynamics. To further improve clarity, please consider (i) briefly reiterating that digital attention and sentiment do not equate to awareness or information quality, and (ii) reinforcing the epidemiological caveat regarding lymphatic filariasis case counts when interpreting attention–burden ratios. These are minor textual clarifications that will enhance interpretability without requiring further analysis.

Reviewer #3: Overall the authors improved their the manuscript but the figures still not in the best quality. some of the figure cannot see the wording and the numbers

PLOS authors have the option to publish the peer review history of their article (what does this mean? ). If published, this will include your full peer review and any attached files.

**Do you want your identity to be public for this peer review?** For information about this choice, including consent withdrawal, please see our Privacy Policy .

Reviewer #1: No

Reviewer #2: No

Reviewer #3: No

**Figure resubmission:**

**Reproducibility:** To enhance the reproducibility of your results, we recommend that authors of applicable studies deposit laboratory protocols in protocols.io, where a protocol can be assigned its own identifier (DOI) such that it can be cited independently in the future. Additionally, PLOS ONE offers an option to publish peer-reviewed clinical study protocols. Read more information on sharing protocols at https://plos.org/protocols?utm_medium=editorial-email&utm_source=authorletters&utm_campaign=protocols

---

## [Editor Report · Decision Letter 2]

17 Feb 2026

Dear Prof. Biswal,

We are pleased to inform you that your manuscript 'Noise and neglect: Social-media signals expose attention gaps for dengue, chikungunya, lymphatic filariasis and kala-azar in India’s vector-borne NTDs' has been provisionally accepted for publication in PLOS Neglected Tropical Diseases.

Best regards,

Shih Keng Loong

Academic Editor

Shaden Kamhawi

Editor-in-Chief

Shaden Kamhawi

co-Editor-in-Chief

Paul Brindley

co-Editor-in-Chief

In light of the authors’ satisfactory responses to all reviewer comments, I recommend acceptance of the article for publication.

---

## [Editor Report · Acceptance letter]

Dear Prof. Biswal,

We are delighted to inform you that your manuscript, "Noise and neglect: Social-media signals expose attention gaps for dengue, chikungunya, lymphatic filariasis and kala-azar in India’s vector-borne NTDs," has been formally accepted for publication in PLOS Neglected Tropical Diseases.

Best regards,

Shaden Kamhawi

co-Editor-in-Chief

Paul Brindley

co-Editor-in-Chief
